# Contribution of Oxidative Stress Induced by Sonodynamic Therapy to the Calcium Homeostasis Imbalance Enhances Macrophage Infiltration in Glioma Cells

**DOI:** 10.3390/cancers14082036

**Published:** 2022-04-18

**Authors:** Lei Chen, Yang Yan, Fangen Kong, Jikai Wang, Jia Zeng, Zhen Fang, Zheyan Wang, Zhigang Liu, Fei Liu

**Affiliations:** 1Department of Neurosurgery, The Fifth Affiliated Hospital of Sun Yat-sen University, Zhuhai 519000, China; chenlei63@mail2.sysu.edu.cn (L.C.); yanyang@mail.sysu.edu.cn (Y.Y.); kongfg@mail.sysu.edu.cn (F.K.); wangjk27@mail.sysu.edu.cn (J.W.); zengj97@mail2.sysu.edu.cn (J.Z.); fangzh56@mail2.sysu.edu.cn (Z.F.); 2The Cancer Center of the Fifth Affiliated Hospital of Sun Yat-sen University, Zhuhai 519000, China; 3Guangdong Provincial Key Laboratory of Biomedical Imaging, The Fifth Affiliated Hospital of Sun Yat-sen University, Zhuhai 519000, China; wangzhy233@mail.sysu.edu.cn; 4Guangdong Provincial Engineering Research Center of Molecular Imaging, The Fifth Affiliated Hospital of Sun Yat-sen University, Zhuhai 519000, China

**Keywords:** grade 4 glioma, sonodynamic therapy, oxidative stress, calcium homeostasis imbalance, macrophage infiltrating

## Abstract

**Simple Summary:**

Sonodynamic therapy (SDT) is a non-invasive technique that is based on the combination of a sonosensitizer and acoustic activation that destroys the mitochondrial respiratory chain, leading to increases in the levels of intracellular reactive oxygen species (ROS) and calcium overload as well as to the inhibition of proliferation, invasion, and promotion of the apoptosis of biologically more aggressive grade 4 glioma. This study aimed to better understand the calcium overload mechanism involved in SDT irradiation and killing gliomas as well as in lipid metabolism in aggressive glioma cells under the SDT treatment. In this study, we examined the hypothesis that the early application of the mechanosensitive Ca^2+^ channel Piezo1 antagonist (GsMTx4) could better promote the dissociation and polymerization of the Ca^2+^ lipid complex and further increase oxidative stress levels, leading to a better anti-tumor effect when SDT was used as a treatment. Moreover, Piezo1’s early closing state and intracellular calcium overload formation may be a key link that leads to the final tumor-infiltrating macrophages.

**Abstract:**

Background: To better understand the Ca^2+^ overload mechanism of SDT killing gliomas, we examined the hypothesis that the early application of the mechanosensitive Ca^2+^ channel Piezo1 antagonist (GsMTx4) could have a better anti-tumor effect. Methods: The in vitro effect of low-energy SDT combined with GsMTx4 or agonist Yoda 1 on both the ROS-induced distribution of Ca^2+^ as well as on the opening of Piezo1 and the dissociation and polymerization of the Ca^2+^ lipid complex were assessed. The same groups were also studied to determine their effects on both tumor-bearing BALB/c-nude and C57BL/6 intracranial tumors, and their effects on the tumor-infiltrating macrophages were studied as well. Results: It was determined that ultrasound-activated Piezo1 contributes to the course of intracellular Ca^2+^ overload, which mediates macrophages (M1 and M2) infiltrating under the oxidative stress caused by SDT. Moreover, we explored the effects of SDT based on the dissociation of the Ca^2+^ lipid complex by inhibiting the expression of fatty acid binding protein 4 (FABP4). The Piezo1 channel was blocked early and combined with SDT treatment, recruited macrophages in the orthotopic transplantation glioma model. Conclusions: SDT regulates intracellular Ca^2+^ signals by upregulating Piezo1 leading to the inhibition of the energy supply from lipid and recruitment of macrophages. Therefore, intervening with the function of the Ca^2+^ channel on the glioma cell membrane in advance is likely to be the key factor to obtain a better effect combined with SDT treatment.

## 1. Introduction

Glioma (CNS WHO Grade 4) is an aggressive and highly lethal cancer. According to the Global Cancer Report 2020, the number of new cancer cases worldwide will reach 28.4 million cases, with 4.1 million new cases being diagnosed by 2040 [1]. Fortunately, the mortality rate has dropped significantly, and the 5-year survival rate has increased slightly [2]. However, most diagnosed patients already have metastases and infiltrations, and surgery often fails to remove all parts of tumor, leading to residual recurrence [3]. Recently, accumulating evidence has suggested that Ca^2+^ is an important positive regulator of early nervous system proliferation and differentiation [4,5]. Additionally, Ca^2+^ has also been found to regulate quiescence, maintenance, proliferation, migration, and cell cycle progression in gliomas [6,7]. Most cancer cells use anaerobic glycolysis to produce ATP under aerobic conditions to obtain higher glucose utilization [8], a phenomenon called the Warburg effect [9]. When there is a lack of intracellular glucose, free fatty acid oxidation can replace glucose to supply energy to cells [10] and can maintain intracellular ROS levels [11] and calcium homeostasis [12]. It seems that Ca^2+^ is closely related to lipid metabolism, so the present study focuses on the role of Ca^2+^ signaling during lipid metabolism.

In deepening our understanding of the interactions between cells, it was determined that when cells are stimulated by an external mechanical pressure, mechanical signals can be converted into biological signals, and this integration is expressed as a cellular response, a phenomenon called mechanical transduction [13]. The mechanosensitive channel on the cell surface converts stimuli into electrical or chemical signals inside cells, and together with the cytoskeleton, they constitute a mechanoreceptor system, which is of great significance to the development of living organisms [14,15]. A new mechanically sensitive ion channel protein (Piezo) was discovered [16] and was found to be able to sense soft touch and harmful mechanical stimuli [17]. In cells, it participates in various physiological processes [18], and in mechanosensing defects in Piezo have been found to lead to various diseases [19]. The trimeric architecture of Piezo1 is more conducive to ion conduction and is determined by high-resolution cryo-electron microscopy [20]. The unique transmembrane helix topology and mechanical transduction components are able to achieve a mechanized mechanism that is similar to a lever [21]. Moreover, Piezo1 expression is closely related to the progression and poor prognosis of glioma and is activated by local mechanical forces to elevate Piezo1 expression to promote glioma aggression [22]. The body senses external mechanical pressure through Piezo1 [23], and high Piezo1 expression has demonstrated good sensitivity and specificity in predicting the degree of glioma tissue edema [24]. Piezo1 regulation results in the formation of spatially structured tissues [25] and manipulates cytosolic Ca^2+^ levels to promote stem-cell proliferation and differentiation [26]. Mechanosensitive ion channels perceive the changes in mechanical stress to regulate cation ion flux, and the cellular Ca^2+^ signaling pathway is needed for homeostasis in brain cancer [27]. It also regulates Ca^2+^ signaling to activate a CAMKK2 AMPK signaling axis for chromosome stabilization [28]. Moreover, high fatty acid transporter (CD36) and fatty acid binding protein 4 (FABP4) expression predicts poor survival and recurrence [29,30]. The complexation of Ca^2+^ and fatty acids seems to be involved in the activation of the mitochondrial apoptosis pathway [31]. Recent studies have shown the great potential of combining microbubbles with the mechanosensitive cation channel Piezo1 after its activation via ultrasound stimulation in sensitizing the response of nerve cells [32]. Additionally, ultrasound stretches the length of the cell membrane, allowing the cell mechanic characteristics to be measured [33]. Although we know that Piezo1 plays an important role in perceiving adverse external stimuli and hardness, there is limited information about the biological effects of Piezo1 when under intracellular oxidative stress after SDT treatment, including the relationship between the Ca^2+^ influx and intracellular lipid metabolism, especially in terms of immune regulation.

In this article, we investigated whether the key role of oxidative stress generated from SDT treatment affected the Piezo1 opening time. Moreover, the Ca^2+^ signaling that combines with fatty acids was then studied by combining Ca^2+^ fluorescence imaging with the ultrasound-induced mechanical loading of glioma cells. Additionally, the positive effects of early changes in Ca^2+^ influx and lipid metabolism on the infiltration of macrophages in glioma cells was determined as well.

## 2. Materials and Methods

### 2.1. Major Reagent and Configuration of Concentration Methods

(1) A 100 mg amount of a photosensitizer that can be activated via ultrasound, i.e., HMME (CAS:148471-91-4, Hematoporphyrinmonomethyl Ether, Yuan Ye, Shanghai, China), that consists of two monomeric porphyrins and that has several desirable properties, such as highly preferential uptake by the tumor and low toxicity to normal cells, was used in the present study. It was dissolved in 10 mL sterile DMEM (Gibco, Thermo Scientific, Waltham, MA, USA) at a ratio of 1:1000 at the final concentration of 10 μg/mL and a pH of 7.4 and was liquefied at room temperature (RT) and stored in the dark at 4 °C; (2) a 1 mg amount of the Piezo1 antagonist GsMTx4 (CAS: TP1300, Topscience, Shanghai, China) was dissolved in 0.047 mL sterile double distilled water at a ratio of 1:1000 at the final concentration of 500 nM and was liquefied at RT and stored in the dark at −20 °C; (3) a 10 mg amount of the Piezo1 agonist Yoda 1 (CAS: T7506, Topscience, Shanghai, China) was dissolved in 2.815 mL sterile DMSO at a ratio of 1:1000 at the final concentration of 10 nM and was liquefied at RT and stored in the dark at −20 °C; (4) a 5 g amount of the ROS antagonist NAC (CAS: CSN16436, CSNpharm, Shanghai, China) was dissolved in 24 mL sterile double distilled water at a ratio of 1:100 at the final concentration of 5 mM, and it was liquefied at RT and stored in the dark at −20 °C; (5) a 50 μg amount of MitoSOX (CAS: M36008, Invitrogen, Waltham, MA, USA), a new type of mitochondrial fluorescent probe that can specifically target mitochondria to selectively detect superoxide in mitochondria, was also used in the present experiment. It was dissolved in 13 μL sterile DMSO at a ratio of 1:1000 at the final concentration of 5 μM and was liquefied at RT and stored in the dark at −20 °C; (6) a 2 mM amount of Flou-4 AM (CAS: S1060, Beyotime, Shanghai, China), a fluorescent probe for detecting the Ca^2+^ concentration in living cells, was also used at a ratio of 1:1000 at the final concentration of 2 μM, liquefied at RT, and stored in the dark at −20 °C; (7) a 10 mg amount of Nile Red (CAS: 19123, Sigma, St. Louis, MO, USA), a selective and hydrophobic fluorescent stain for intracellular lipid droplets and neutral lipids with living cells, is intensely fluorescent in all organic solvents, and the fluorescence colors range from golden yellow to deep red [34]. It was dissolved in 3.141 mL sterile DMSO, had a ratio of 1:10,000 at the final concentration of 1 μM, and was liquefied at RT and stored in the dark at 4 °C; (8) a 0.5 mL amount of Hoechst 33342 (CAS: C1028, Beyotime, Shanghai, China), a blue fluorescent dye solution that stains the nucleus of living cells and that can bind to double-stranded DNA. When a cell undergoes apoptosis, it densely stains the nucleus, or it may be fragmented and densely stained. This was added into the culture medium at a 1:1000 ratio, was liquefied at RT, and stored in the dark at −20 °C.

### 2.2. Glioma Cell Cultures

Human glioma cell lines (U251 and U87) and a mouse glioma cell line (GL261) were obtained from the National Cancer Institute and ATCC (American Type Culture Collection). U251 and GL261 were carefully cultured in DMEM (Gibco, Thermo Scientific, MA, USA) consisting of 10% fetal bovine serum (Gibco, Thermo Scientific, MA, USA) and 1% penicillin/streptomycin (Thermo Fisher Scientific, Waltham, MA, USA) and U87 with RPMI 1640 (Gibco, Thermo Scientific, MA, USA). All cell lines were maintained at 37 °C with 5% CO_2_.

### 2.3. Ultrasound Generator System

The experimental setup is shown in Figure 1A. We used an adjustable frequency ultrasonic generator (each group uses 6 holes/96 plate) to carry out the CCK8 experiment to obtain better time and frequency parameters (<IC50). In this work, we used a physical therapy ultrasound device (JP-010T, Shenzhen, China) with two probe planar (d = 14 cm or d = 3.5 cm) ultrasound transducers that could be adjusted to generate ultrasound waves with a 40 kHz frequency and 0.5 W/cm^2^ power density. As per our previous experimental method [35], power (W) = (dT/dt) cpM, cp is the heat capacity of water (4.2 J g^−1^), M is the mass of water (g), and (dT/dt) is the temperature increase per second [36]. The top of the probe planar is big enough for the whole bottom, and a 1 cm thick sponge soaked with double distilled water (temperature at 24 °C) was placed there to reduce heat conduction and was replaced after each step. Twelve groups were formed: the control group (CON), ultrasound group (US), HMME + US group (SDT), US + GsMTx4 group (UG), US + Yoda 1 group (UY), US + NAC group (UN), SDT + GsMTx4 group (SG), SDT + Yoda 1 group (SY), SDT + NAC group (SN), GsMTx4 group (G), Yoda 1 group (Y), and NAC group (N). After ultrasound treatment, the above drugs were added to their respective groups and tested.

### 2.4. Cell Survival Rate

While in the logarithmic growth phase, 3 × 10^3^ U251 and U87 cells/holes were digested using trypsin and seeded in 96-well plates (six holes/group/plate). For the SDT, HMME (final concentration of 10 μg/mL) was cultured in the dark for 2 h before the ultrasound treatment and was tested using Cell Counting Kit-8 reagent (CCK-8, Dojindo, Kumamoto, Japan) to determine the experimental parameters and the application time point, which was carried out using a previously described experimental method [35]. Then, the absorbance at 450 nm was measured at three time points (1, 2, and 3 h) and was tested using an enzyme mark instrument (MOLECULAR DEVICES CMax Plus, SiliconValley, Austin TX, USA).

### 2.5. Intracellular Ca^2+^ with Flow Cytometry

Intracellular Ca^2+^ signals were measured with Fluo-4 AM (CAS: S1060, Beyotime, Shanghai, China) according to the manufacturers’ instructions. In detail, cells were incubated with Fluo-4 AM (final concentration of 2 μM) for 30 min in PBS at 37 °C and were then washed three times with PBS and incubated for an additional 20 min to completely transform them into Fluo-4 inside cells. The fluorescence intensity was tested using a flow cytometer (CytoFLEX LX, Brea, CA, USA).

### 2.6. Fluorescence Method for Intracellular Ca^2+^/Lipid Co-Expression

To measure the fluorescence of the intracellular Ca^2+^ from the U251 cells 24 h after the different treatments, the cells were incubated with Fluo-4 AM at a final concentration of 2 μM and were incubated in a Forma Series II Water Jacketed CO_2_ incubator (3111, Thermo Scientific Forma, Waltham, MA, USA) at 37 °C with 5% CO_2_ for 1 h and then washed twice with Hanks Balanced Salt Solution without calcium magnesium and phenol red (1 × HBSS, PH1511, Phygene, Fujian, China), pH 7.4, and was incubated for another 30 min and then tested. Then, Nile Red (1 μM) was added to the confocal dishes, incubated at 37 °C with 5% CO_2_ for 10 min, washed twice with HBSS, and tested using a confocal microscope (ZEISS LSM 880, Oberkochen, Germany).

### 2.7. Piezo1 with Ca^2+^ or MitoSOX Was Detected via the Immunofluorescence Labeling Method

Different treatments for the U251 cells were applied according to the manufacturer’s protocols after 24 h. First, the intracellular Ca^2+^ with Fluo-4 AM was fluorescently stained and then incubated with MitoSOX (5 μM) at 37 °C for 10 min in the dark, washed twice with Hanks with calcium magnesium (1 × HBSS, PH1509, Phygene) pH 7.4, and pre-cooled at 4 °C in 4% paraformaldehyde and fixed for 10 min at RT. Other U251 cells were stained with Fluo-4 AM and then blocked with 10% normal donkey serum (G1217, Servicebio, Wuhan, China) for 30 min at RT and were then incubated with rabbit anti-Piezo1 antibody (1:200, 15939-1-AP, Proteintech, WuHan, China) at 4 °C overnight (about 16 h). The next day, the fluorescence-labeled secondary antibodies were added (1:400, 711-545-152, Alexa Fluor 488-ChromPure Donkey IgG, Jackson ImmunoResearch, West Grove, PA, USA; according to the instructions, 0.5 mg was rehydrated with 0.4 mL ddH_2_O at an antibody concentration 1.5 mg/mL) for 1 h at RT in the dark. Finally, they were washed with PBS three times (5 min each time) and were then incubated with DAPI (G1012, Servicebio) at RT in the dark for 15 min and then washed three times again. The cell nucleus was stained with Hoechst 33342 (CSA: C1028, Beyotime) for 10 min at 37 °C in the dark, and images were captured with a confocal microscope (ZEISS LSM 880, Oberkochen, Germany) (×40 oil immersion lens).

### 2.8. The Dynamic Changes in Ca^2+^ and Lipids in Living U251 Cells

U251 cells (2 × 10^4^/well) were incubated in confocal dishes, and the next day, after the different treatments had been applied for 1 min, the cells were put under a microscope for real-time observations using a living cell analysis system (Axio Observer 7, ZEISS) at 37 °C with 5% CO_2_, and observations continued for 10 min after the addition of the different drug groups.

### 2.9. Western Blot

All experimental groups were then treated for 3, 6, 12, and 24 h. The cells were collected and washed before lysis. A total amount of 30 μg of proteins and 15% Tris–glycine gels (Solarbio, Beijing, China) were isolated in the SDS-PAGE, transferred to PVDF membranes (Millipore, Bedford, MA, USA), and then incubated with the rabbit anti-FABP4 antibody (1:1000, 12802-1-AP, Proteintech) at 4 °C overnight (about 16 h). Then, rabbit secondary antibodies (Affinity Biosciences, Jiangsu, China) were added to the membranes, which were then diluted to 1:5000 and incubated for 1 h at RT. They were then cleaned with 1 × PBST buffer three times, and exposed by the SAGECREATION, and each experiment was repeated at least three times. Finally, the membranes were analyzed using Image J software.

### 2.10. Animal Models and Treatment Regimes

Male BALB/c nude mice and male C57BL/6 mice (4 weeks old and weight > 16 g) were purchased from the Guangdong experimental animal center (Guangdong, China). The Animal Ethics Committee of the Fifth Affiliated Hospital of Sun Yat-sen University approved the animal experiments (NO.00124). The mice were raised in the Animal Barrier System at the Guangdong Provincial Key Laboratory according to the protocols approved by the committee. All animal experiments were conducted in line with the Guide for the Care and Use of Laboratory Animals. After 1 week of raising the BALB/c-nude mice (those weighing between 17 and 19 g were selected for the experiment), the nude mice were subcutaneously injected into their right lateral thighs. Subcutaneous tumor growth was observed every other day and was measured with vernier calipers. When the subcutaneous tumor volume reached about 100 mm^3^, the nude mice that met the requirements were randomly divided into four groups (n = 5/group): control group (PBS); HMME + US group (SDT); SDT + GsMTx4 group (SG); SDT + Yoda 1 group (SY). The HMME injection dose (100 mg dissolved with 10 mL PBS) was 0.05 mL/10 g body weight, and the mice were kept out of the light for 12 h before undergoing ultrasound. The above SDT, SG, and SY groups received ultrasound irradiation every other day (3 times) and had 1 μL GsMTx4 or Yoda 1 injected into the subcutaneous tumor after irradiation (3 times). After SDT treatment, the nude mice were protected from light for 48 h to avoid skin photosensitivity. We assessed the tumor volume every other day (followed the formula V = L × W^2^/2, L, length, W, width) as well as the body weight of every mouse. A total of 23 days after the treatment was first administered, we obtained the remaining subcutaneous tumors, and the left ventricle was perfused with 4% paraformaldehyde to be able to fix it for further testing later. The C57BL/6 mice were anesthetized with isoflurane gas (300 mL/min) while fixed in the prone position. After incising the skin, the anterior fontanel was exposed and moved back and to the right by 2 mm. A 1.0 mm hole was drilled while ensuring that the dura was not punctured, and 5 μL of the cell suspension was injected with a reversed micro syringe at a 3 mm depth for 3 min before the needle was removed slowly, and the skin and fascia were then sutured. After the operation, the temperature was kept at 37 °C until the mice woke up. Weight changes continued to be monitored daily, and weight loss was determined to be more than 1.0 g/day as the tests and experiments continued.

### 2.11. The 9.4 T MRI Detection

The orthotopic gliomas from the brains of male C57BL/6 mice were tested by 9.4 T MRI at 0 and 24 days after the transplantation of orthotopic GL261 cells. First, the metal ear tags on the mice were removed, and then 400 mL/min isoflurane was used to anesthetize the mouse. The anesthesia state was maintained with 150 mL/min isoflurane/air mixed gas for 10 min, and the mice were fixed to the scanning bed with a mouse body coil and were connected to the ECG gating and respiratory gating devices. Importantly, the heating pad under the abdomen and the changes in the breathing and heartbeat were monitored throughout the whole process, with the number of breaths stabilized at 40–60 times/min, and the heart rate stabilized at 100–150 times/min. Scanning was carried out with the 9.4 T magnetic resonance scanner (BioSpec 94/30 USER, Bruker Biospin MRI Gmbh, Ettlingen, Germany). The imaging sequence and parameters were as follows: gradient echo sequences T1 Flash (TR 283.698 ms, TE 3 ms, FOV 20 mm × 20 mm, matrix 384 × 384, thickness 0.7 mm); fast spin-echo sequences T2 TurboRARE (TR 1550.0 ms, TE 23 ms, FOV 20 mm × 20 mm, acquisition matrix 320 × 320, thickness 0.7 mm). Magnevist (200 μL/kg, 20 mL, 9.38 g, Schering Pharmaceutical Ltd., Ettlingen, Germany) was injected into the tail vein of the mouse within 5–10 min before the enhanced MRI scan was carried out.

### 2.12. Immunofluorescence Double Staining of Brain Orthotopic Gliomas from the C57BL/6 Mice

At the end of the animal experiments, all experimental C57BL/6 mice were sacrificed, and the brains were completely removed and were perfused with 4% paraformaldehyde and were immersed for one week and then dehydrated with gradient sucrose for 48 h, embedded in optimal cutting temperature compound (OCT) (SAKURA, CA, USA), frozen and sectioned into portions that were 35 μm thick, and stored in antifreeze buffer for testing. The brain slices were taken from the antifreeze buffer and washed twice with PBS and then blocked with 10% normal donkey serum (G1217, Servicebio) for 1 h at RT followed by incubation in rabbit anti-CD86 antibody (1:200, DF6332, Affinity) and rabbit anti-MRC1 (CD206) antibody (1:200, DF4149, Affinity) at 4 °C overnight (about 16 h). The next day, the fluorescence-labeled secondary antibodies (1:400, 711-545-152, Alexa Fluor^®^ 488-ChromPure Donkey IgG, Jackson ImmunoResearch) were added for 2 h at RT in the dark. Finally, they were washed with PBS three times (5 min/time) and were incubated with DAPI (G1012, Servicebio) at RT in the dark for 15 min and then washed three times again. Images were captured with a confocal microscope (ZEISS LSM 880, Oberkochen, Germany) (×40 oil immersion lens).

### 2.13. Hematoxylin and Eosin (H&E) and Immunohistochemical Staining

After the fixed tumor tissue was embedded in paraffin, tissue was sectioned into very thin (5–10 μm) sections using a microtome, and these paraffin sections were placed on slides and kept warm in an oven at 60 °C for 2 h. Then, the sections were deparaffinized with xylene for 15 min, and the tissue was successively rehydrated with 100%, 90%, 80%, and 70% alcohol for 5 min and stained with hematoxylin and eosin for 2–3 min (C01055, Beyotime, Shanghai, China), and the tissue was then dehydrated successively with 80%, 90%, and 100% alcohol and finally with xylene for 5 min. Finally, the neutral resins were mounted and left to dry overnight to create permanent slides. The primary antibodies rabbit anti-Ki 67 antibody (1:200, 27309-1-AP, Proteintech, WuHan, China) and rabbit anti-Caspase 3 antibody (1:200, 19677-1-AP, Proteintech, WuHan, China) were incubated overnight according to the instructions of the immunohistochemistry kit (SV0002, BOSTER, Wuhan, China) used in this study. The next day, the secondary antibody was incubated for 30 min and then DAB stained and observed under the microscope so that pictures could be taken.

### 2.14. Statistical Analysis

The data were expressed as mean ± SD. Statistical comparisons between groups were evaluated by Student’s *t*-test or one-way ANOVA followed by Dunnett’s multiple comparisons test, which was performed using GraphPad Prism version 8.0.0. Fluorescence data were analyzed using confocal analysis (ZEN Blue Lite) and Image J software. All experiments were performed in triplicate. * *p* < 0.05, ** *p* < 0.01, and *** *p* < 0.001 were considered statistically significant.

## 3. Results

### 3.1. SDT Inhibits the Proliferation of U251 and U87 Glioma Cells

In order to find better experimental parameters for SDT, the rate of inhibition on U251 and U87 glioma cell proliferation was determined by the CCK8 method at different exposure frequencies (0.25, 0.5, 1.0, 1.5, 2.0, and 2.5 MHz) (Figure 1B,C). At 1.0 MHz, the inhibition rate was close to IC50, so a smaller frequency of 40 kHz was used with a lower inhibition rate (<20%) which we were able to adjust using the equipment used in the study (Figure 1A). Additionally, SDT treatment demonstrated a better action time with the 40 kHz frequency (60, 120, 180, 240, 300, 360, 420, and 480 s), and we chose 60 s as the acting time for the following experiments (Figure 1B,C).

### 3.2. SDT Interferes with Intracellular Ca^2+^ Homeostasis via Increasing Mitochondrial Oxidative Stress Levels

The oxidative stress level of the intracellular mitochondria was detected by MitoSOX immunofluorescence. Firstly, the flow cytometer results illustrated that there were obviously different Ca^2+^ levels between the SDT treatment groups and other groups in the U251 cells (*p* < 0.01), and these levels were higher in the SDT group than they were in the SG and SN groups (Figure 1D,E). However, in the U87 cells, the levels were higher in the SG group than they were in the SDT and SY groups (Figure 1F,G). In the SDT group, ROS were not only present in the mitochondria, but also entered the nucleus (Figure 2A). The oxidative stress level and fluorescence intensity of Ca^2+^ in the CON and US groups were both low level (Figure 2B). However, both the oxidative stress level and Ca^2+^ were greatly increased in the SG group (Figure 2A,B), and there was a concomitant increase in death in the SG group (Figure 2A). In contrast, the level of intracellular ROS in the SY group was lower than it was in the SDT and SG groups, but the level of intracellular Ca^2+^ in those groups was higher than the intracellular Ca^2+^ level in the SDT group (Figure 2A,B). Importantly, the intracellular MitoSOX level was the lowest in the SN group, and intracellular Ca^2+^ was lower in the SN group than it was in SG and SY groups, but higher than it was in the SDT group (Figure 2A,B).

### 3.3. SDT Regulates the Redistribution of Lipid Droplets and the Ca^2+^ Lipid Complex by Affecting Ca^2+^ Homeostasis

Figure 3A shows fluo-4 AM and Hoechst 33342 staining under microscopy. The US and SDT groups showed Ca^2+^ influx and local accumulation on the cell membrane, and the SDT group demonstrated obvious damage in the nucleus (nucleus was stained navy blue and nuclear fragmentation) 24 h after SDT treatment (Figure 3A). In the SG, SY, and SN groups, the nuclei were deeply stained, but the Ca^2+^’s fluorescence intensity was lower than that of the US and SDT groups (Figure 3A,C). At the same time, the SDT group had a significantly reduced lipid droplet content compared to the CON and US groups, and the lipids were also reduced in the SN group (Figure 3B); however, a more significant reduction was observed in the SG group (Figure 3B,D). The Ca^2+^ that had co-localized with the lipid droplets was present in all groups (r > 0.80) (Figure 4A). Additionally, the colocalization coefficient was analyzed using the Pearson correlation coefficient (PCC) (Figure 4C) and the Manders overlap coefficients (MOC) (Figure 4D). They showed that the co-localization in the SG group was lower than that in the SDT group (Figure 4C,D). We measured the temperature changes before and after ultrasound treatment using a thermometer and changed double distilled water every time (to maintain temperature fluctuations within 2 °C) (Figure 4F). The U251 cells received SDT treatment (Appendix A) at 3 and 24 h, and it was observed that the cells were swollen and that the Ca^2+^ located on the cell membrane entered those cells. The 24 h pictures were overexposed so that the swollen outline of the cell membrane could be seen clearly (Appendix A). Additionally, the short video showed that the CON + GsMTx4 (CG) group had better membrane fluidity, expansion, and contraction than the SG group did, which presented a lack of elasticity and was motionless (Appendix A).

### 3.4. SDT Maintains the Opening State of Piezo1 Channels by Increasing Intracellular Oxidative Stress Levels

Figure 4B shows the fluorescence intensity of the changes that take place in intracellular Ca^2+^ and Piezo1 expression under different treatment conditions. In the SDT group, the fluorescence of intracellular Ca^2+^ and the Piezo1 expression were higher than they were in the CON and US groups. Under SDT, the Piezo1 expression in the SG 396 group was downregulated, but the intracellular Ca^2+^ was significantly increased. After the Piezo1 channel was inhibited, the calcium ions that enter the cell could not flow out of the cells, resulting in an increase in intracellular Ca^2+^, and the nucleus was fragmented and not clear. In contrast, the Piezo1 expression in the SY group was significantly more enhanced than it was other groups, but the intracellular Ca^2+^ decreased. The Piezo1 channel is excessively opened, and intracellular calcium ions flow out of the cell along the concentration difference. Importantly, both the expression of Piezo1 and intracellular Ca^2+^ were decreased in the SN group (Figure 4B,E). Inhibition of reactive oxygen species reduces mechanical stimulation of cells, resulting in a decrease in Piezo1 expression. A short video showed that the vesicles that protruded outside of the cell membrane did not shrink or disappear after the addition of GsMTx4 leading to intracellular Ca^2+^ being locked, indicating that the cells were already in a state of irreversible death after SDT treatment (Appendix A).

### 3.5. SDT Prolongs the Piezo1 Opening Time and Affects the Ability of Ca^2+^ and Lipid Droplets to Enter Cells

Next, the opening time course of the Piezo1 channel and the effect of the antagonist and agonist on Ca^2+^ influx and the distribution of intracellular lipid droplets after treatment was administered to the CON, US, and SDT groups were determined. Figure 5A shows that after one minute of sonication treatment, the Ca^2+^ fluorescence intensity (fi) of the CON group gradually decreased and was maintained at 600 fi. However, the Ca^2+^ fluorescence intensity (fi) of the US group gradually decreased and remained at a low level (460 fi) (Figure 5C). On the contrary, the SDT group showed a strong upward trend (up to about 760 fi), which dropped sharply at 100 s and gradually decreased after 200 s and was maintained at 600 fi, which was the same level observed in the CON group (Figure 5E); within the next 10 min from the addition of GsMTx4, the UG (1350 fi) and SG (4000 fi) groups had a brief high peak and gradual decrease (Figure 5C,E). GsMTx4 was added at the 600 s time point, and through the living cell analysis system, it was observed that the three groups showed different degrees of decline. In the same microscopic field, the continuous Ca^2+^ fluorescence intensity changes within a single cell were analyzed over the course of 10 min. Nile red was used to stain the lipid droplets in another group of U251 cells that had received the same treatments, and the initial 10 min images showed and upward trend for the increasing fluorescence intensity. Within the next 10 min from the addition of GsMTx4, the SG group decreased evenly, and all the three groups temporarily plateaued at 600–800 secs, and the CG and UG groups showed an upward recovery trend (Figure 5F). The sudden ascent of the UG group was different from the zigzag downward trend of the CG group (Figure 5B,D). Western blot experiments showed that SDT had already started to inhibit the expression of FABP4 at 3 h, and this continued to 12 h (Figure. 6A). The data from the U251 and U87 cell lines are slightly different. The SG group showed the strongest effects at 3–6 h in both cell lines (Figure 6A–C). In the U87 cell line, these effects occurred at 3 h, earlier than the 6 h required for them to be observed in the U251 cell line (Figure 6B,C). Additionally, at 12 h, the ROS inhibitor NAC (SN group) showed the strongest effect, which is in contrast with the SDT group (Figure 6B,C).

### 3.6. GsMTx4 Can Enhance the Effect of SDT and Promote the Macrophages Infiltrating in Glioma

According to the schematic diagram of the animal experiment shown in Figure 7A, the subcutaneous tumor-bearing BALB/c nude mice were treated with CON, SDT, SG, and SY, with the treatment to be received being determined based on groups. After adding 1 μL GsMTx4 or 1 μL Yoda 1 to the subcutaneous tumors in the nude mice, the tumor volume (Figure 7B,C) and weight (Figure 7D) were significantly suppressed. The tumors in the SG group were smaller than those observed in the other groups. Pathological sectioning was performed with immunohistochemical staining, and in the SG group, Ki67 was significantly reduced, Caspase-3 was significantly increased, and there were multiple necrotic areas that could be observed after H&E staining (Figure 7E). Additionally, in the C57BL/6 mice, 9.4 T MRI confirmed the formation of gliomas in the CON group (a daily weight loss rate of less than 1.0 g was selected) (Figure 8A), and when the other groups started SDT treatment, the right brain ventricle was punctured and injected with 2 μL each. At 30 days, frozen brain sections were created for dual fluorescence staining. During SDT treatment, CD86 (M1 marker) gradually increased, and the expression in the SG group was the higher than it was in the SDT and SY groups, with the lowest being observed in the CON group (Figure 8B,C). Similarly, the CD206 (M2 marker) expression in the other groups was higher than it was in the CON group. Interestingly, its expression in the SDT group was higher than it was in the SG and SY groups (Figure 8D,E).

## 4. Discussion

Although a great deal of progress has been made in the development of various treatment methods for patients with grade 4 glioma, the 5-year relative survival rate is only 23.5% [37]. Moreover, deep infiltrating glioma cells cannot be completely removed, and resistance to chemotherapeutics and radiotherapy are still the main reasons for recurrence and death [38]. Indeed, subsequent PDT therapies provide a glimmer of hope for grade 4 glioma patients, but recently it was discovered that there are limitations that affect the penetration depth of these treatments, meaning that they are unable to effectively kill infiltrating glioma cells deep in the brain [39]. The discovery of SDT has made up for this defect, but there are also many unsolved problems [40]. The studies presented here are the first to imply that ultrasound-activated Ca^2+^ signals block the lipid metabolism of glioma cells. SDT is able to mechanically stimulate cells with the assistance of ultrasound, and at the same time, it was also able to generate more ROS. Moreover, when Piezo1 was kept open, more calcium ions were able to enter the cells and combine with the lipid droplets to form a complex, affecting the lipid metabolism and preventing it from providing energy for glioma cells. On the one hand, the mechanical force introduced by the ultrasound activated the opening of the transient calcium ion channel Piezo1 on the membrane, increasing Ca^2+^ influx, and the cell swelled and activated the death mechanism known as the calcium overload pathway. On the other hand, the ROS that were produced by the intracellular sonosensitizers that were activated by ultrasound prolonged the Piezo1 opening time, and the excessive Ca^2+^ promoted intracellular lipid peroxidation and lost the ability to supply energy to the cells, resulting in cell death (Figure 1). Out of all studied treatments, SDT played a significant role in initiating the expression and function of the transient mechanically sensitive calcium channel Piezo1 in glioma cells. Moreover, we observed that the window of time in which the calcium ion channel Piezo1 was open and intracellular lipid deficiency provided more important evidence of the preliminary experimental basis for regulating Ca^2+^ channels or for lipid metabolism target drugs for SDT in vitro and vivo.

High Piezo1 and Piezo2 activity plays an essential role in many physiological processes, including touch and pain sensations, hearing, and blood pressure regulation, because of their ability to perceive mechanical forces [16]. Several works have been dedicated to the gating mechanisms of Piezo1 channels, which form a trimeric propeller-like structure [20]. Piezo1 comprises mechanically gated cation channels [16], and TRP family channels are also known to be another leading candidate for mechanosensitive channels, but Piezos can largely function on their own without relying on others [41]. The structure of the Piezo1 protein [20,42] and the mechanized mechanism [21] were successfully interpreted, and it was found that intracellular Ca^2+^ acts as a regulatory messenger effector that plays an important role in cell life activities [43]. More attention should be focused on whether a more structured mechanical microenvironment elevates Piezo1 expression to promote glioma aggression and tumor development [22]. Piezo1-mediated mechanotransduction can be applied in nervous system pathologies and in solid cancers, including in mesenchymal gliomas, which possess more treatment resistance and invasiveness. It is able to successfully sense and quickly respond to the surrounding microenvironment and increase the stiffness of the extracellular matrix (ECM) and induce a mesenchymal-like phenotype in tumors, potentially making Piezo1 a viable therapeutic target for treatment-resistant and invasive gliomas. Just like SDT treatment, the pressure caused by the mechanical vibrations and the stimulation induced by oxidative stress act as a double blow, as the pathway is harmful to cells when left open. Moreover, Piezo1 is involved in focal adhesion assembly and in the activation of the integrin-FAK pathway, providing a mechanistic explanation for its anti-tumor activity [44]. Piezo1 overexpression may become a new prognostic biomarker for glioma patients [45,46] and may be able to predict the edema extent of glioma tissues [24].

Piezo channels are a critical regulator of glioma angiogenesis via Ca^2+^ signaling [46,47]. The dysregulation of forces in the cell and tissue mechanics can activate mechanosignaling to compromise tissue integrity and function and promote brain cancer progression [48]. Recently, the effect of different types of pressure on the Piezo1 activation via ultrasound were determined [32], but we emphasized the importance of oxidative stress in promoting the function of the Piezo1 channel. During SDT treatment, increased Piezo1 expression is accompanied by increased intracellular Ca^2+^, and more Ca^2+^ will be trapped inside the cell due to the effects of GsMTx4. Yado1 augmented intracellular Ca^2+^ oscillations, but it did not elevate subsequent Ca^2+^ influx, and our data were consistent with previous studies [49]. From Figure 3 and Figure 4A, the intracellular lipid accumulation and redistribution on the membrane was not only caused by the reduction in Ca^2+^ influx, but the oxidative stress level was also the result of SDT, which, based on our results, plays a key role in the homeostasis imbalance of Ca^2+^. The ROS that was generated by SDT can activate the mitochondrial apoptosis pathway [35], leading to mitochondrial dysfunction and DNA damage, keeping MCU channel activity persistent, inducing Ca^2+^ overload in the mitochondria, further aggravating mitochondrial dysfunction, generating more ROS, and forming a vicious cycle that ends in death [50]. US irradiation simulates pressure changes in the tumor microenvironment and the instantaneous cation channel Piezo1 is observed to open, resulting in an influx of Ca^2+^. We applied SDT to apply a smaller amount of pressure (20 µPa^40^ kHz = 800 µPa), and this also activated Piezo1 even though the sound pressure value was lower compared to previous studies (0.03 MP) [31]. Moreover, SDT prolongs the peak time of the intracellular Ca^2+^ level (to 200 s) compared to in previous works (40–80 s at a high level) [32], meaning that other treatment therapies have sufficient time for combination therapy to take place. At the same time, the ROS interfered with the mitochondrial oxidative respiration function, allowing more Ca^2+^ to enter the mitochondria, leading to mitochondrial calcium overload and dysfunction. The disturbance of Ca^2+^ homeostasis and mitochondrial abnormalities in the pathophysiological process represent early changes in a complex disease [51]. Previous studies focusing on SDT and calcium overload interactions emphasized that the release of Ca^2+^ from the effector organelles into the cytoplasm is the key step in the mitochondrial apoptosis pathway [52]. Our results showed that not all the glioma cells with Ca^2+^ influx induce cell swelling and apoptosis. Moreover, GsMTx4 was able to reverse the accumulation of intracellular lipid droplets by blocking more Ca^2+^ from entering the cells and preventing the intracellular Ca^2+^ from being exchanged with the external environment. The influx of Ca^2+^ resulted in the cells having a new destiny, one in which it is easier to be killed [6]. Moreover, the Ca^2+^ was involved in regulating blood–brain barrier (BBB) permeability by activating transient receptor potential vanilloid 4 (TRPV4) [53]. There were local vesicles around cells after 24 h, indicating that there is a recovery transformation process, and then swelling at 3 h and death occurred. All of the above provide valuable information, and when the cancer cells were less stiff, it was easier for them to react to US treatment, and lower stiffness made energy transfer to the inside of the cells easier [54]. Our initial focus on the cell membrane lipid particles was based on strong evidence of the co-localization and dissociation of Ca^2+^ and lipid droplets. The changes in plasma membrane fluidity can result in the cells being swollen during SDT treatment, and the mechanism of the phenomenon is worthy of further study.

The Piezo1 channel also participates in many other physiological activities, such as erythrocyte volume homeostasis [55], correct pathfinding during axonal growth [56], brain vascular development [57,58], and complete aortic valve development in zebrafish [59]. Based on the above, Piezo1 could find ways to maintain life-related activities by sensing changes in the surrounding environment, and a lack of Piezo1 would lead to a variety of physiological obstacles and even diseases. Moreover, Piezo1 can mediate the extrusion of overcrowded living cells and the growth and division of epithelial cells in sparse areas [18], something that can be attributed to sensing both mechanical crowding and stretching in a steady state [60], and it could also act as a homeostatic sensor for Ca^2+^ to regulate the number of glioma cells. Decreased membrane fluidity with significant cellular surface deformation could be attributed to a sono-chemical activation mechanism [61]. The imbalance of intracellular homeostasis led to the declined membrane fluidity and cell swelling, which was perhaps the key activating the effects impaired by SDT. Compared to tumors, Piezo1 can maintain life activities in the steady state, but in our study, SDT caused a homeostasis imbalance of Ca^2+^, which led to mitochondrial dysfunction due to Piezo1 activation. Moreover, membrane fluidity can be altered by the local pressure variations caused by ultrasound exposure and show a time-dependent decreasing trend [62]. The sonicated cells exhibited membrane swelling and intracellular lipid consumption at 24 h, which was different from the results at 2 h (membrane shrinkage and intracellular lipid accumulation) [63]. The ROS inhibitor protects the cell membranes from oxidative injuries to improve membrane fluidity [64]. Ultrasonic cavitation activates the calcium channels on the membrane, which deliver the signals to the intracellular mitochondria through membrane lipid fluid, and mitochondrial calcium uniporter (MCU) inhibition due to mitochondrial dysfunction, preventing mitochondrial calcium uptake and promoting M1 polarization [65]. However, how spatial confinement and the stiffness of the in vitro environment co-regulate the polarization of the macrophages as well as how Piezo1 mediates calcium influx macrophage polarization regulation needs further research [66]. Preliminary data show that the early-stage intervention in Ca^2+^ channels combined with SDT results in an increased proportion of M1 macrophage infiltrating. However, in the SDT group, increased M2 infiltration was found in the detection at 30 days, which may be a potential factor for glioma recurrence after SDT treatment [67]. Unfortunately, there is no more convincing data on the detection of macrophage cytokines as well as the detection of macrophage infiltration in shorter time periods, such as 3, 7, and 15 days. The opening of the Piezo1 calcium channel is a dynamic process that may change throughout tumor development. High ROS levels stimulate tumor cells to adapt to oxidative stress by increasing metabolism through the pentose phosphate pathway (PPP) [68] or turn into lipid energetic pathways to maintain the malignant state [69]. The SDT treatment not only generated more ROS, but also blocked the function of FABP4 protein in lipid metabolism. Although we found that the two glioma cell lines had different inhibitory effects on FABP4 at different time points, significant inhibitory effects could be seen at 3, 6, and 12 h. The two cells were wild type and mutant on P53, respectively, the sensitivity to ROS induced by SDT treatment is different, and the regulatory pathway of P53 may also be involved in its lipid metabolism [70]. In addition, at 3 h in the SN group of U251 cells, there was a significant upregulation of FABP4, which may be due to the activation of FABP7 to form lipid droplets against oxidative stress [71].

We will further explore the following scientific questions regarding non-lethal sonodynamic therapies: On the one hand, which calcium channels are involved in providing calcium for intracellular calcium overload and for intervention during intracellular lipid transport and oxidation during different development stages? On the other hand, what are the effects of SDT intervention on the relationship between the ability of the macrophages to uptake calcium and the metabolite microenvironment in glioma cells? Despite some limitations, we verified the biological effects after 24 h but were unable to verify the biological effects at shorter time (3 and 6 h) points. Moreover, the effects of dynamic changes in the ROS on the expression and function of Piezo1, and macrophage infiltration require further research.

## 5. Conclusions

We demonstrated that SDT activated the opening of the transient calcium channel Piezo1 to regulate intracellular Ca^2+^ homeostasis in response to lipid homeostasis imbalances caused by oxidative stress. Increasing the level of intracellular oxidative stress further aggravated the Ca^2+^ fluctuations and induced mitochondrial calcium overload, leading to mitochondrial dysfunction. Insufficient ATP supply and lipid metabolism disorders were caused by environment-sensing malfunctions. Taken together, our results suggest that SDT regulates intracellular calcium signals by upregulating Piezo1, by cooperating with increasing ROS, and by inhibiting glioma cells from becoming the energy supply chain for lipid metabolism under oxidative stress.

## Data Availability

The data used and analyzed during the current study are available in the manuscript text and tables. Further information could be obtained from the corresponding author on reasonable request.

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
