# Peer review of "Contribution of Oxidative Stress Induced by Sonodynamic Therapy to the Calcium Homeostasis Imbalance Enhances Macrophage Infiltration in Glioma Cells"

_cancers, 2022, doi:10.3390/cancers14082036_

Round 1
Reviewer 1 Report
The authors have sufficiently addressed all of my concerns for this manuscripts and have corrected the figures and English language issues. They have also discussed the potential heterogeneity between various glioma cell lines in reference to their study.
Author Response
We greatly appreciate your review and encouragement, adding confidence to our future calcium channel research. Thank you again for your suggestion, and learn more from you.
Please see the attachment.

Reviewer 2 Report
The authors present an interesting study determining whether the early application of Piezo1 could augment the anti-tumor effects of sonodynamic therapy in glioma model.
Here are a few suggestions:
Major comments:
- The title and the results say that there is M1 polarization. However, the authors have tested only 2 markers, CD86 (M1) and CD206 (M2) marker and both are up regulated in their model. Since, this is one of their key points, it is essential that they test additional M1 markers to support their claim of M1 polarization.
https://www.frontiersin.org/articles/10.3389/fimmu.2017.01097/full
https://www.ncbi.nlm.nih.gov/pmc/articles/PMC5566322/
Minor comments:
- The authors have also seen and reported differences post the various treatments in the two cell lines, particularly in their SN group. For example, Figure 6, Line 430; The authors need to discuss the potential reasons for these variations. This is important because the authors are using only 2 cell lines for their experiment, so it is essential that the data match between the two for it to be a universally applicable finding.
- The authors need to explain their results more. For instance, in section 3.4, what do the differences in the Piezo1 and intracellular Ca2+ between the groups signify?
Round 2
Reviewer 2 Report
Hi,
The authors have improvised their work and the revised version reads much better. However, a major concern still remains. The current data does not support the the claim of M1 macrophage polarization, which is included in their title too. While the upregulation of CD86+ cells in SG group is significant, it is inadequate to state that there is M1 polarization. at least 1 other marker for M1 marker is required. This is important because the authors have shown an upregulation of a M2 marker CD206 as well. This is particularly important because the authors have data only from one (GL261 derived) tumors.
Round 3
Reviewer 2 Report
The revised manuscript read much better and I recommend acceptance.
Just a suggestion : If in case you wish to retain macrophages in your title, you may rephrase it to "Contribution of oxidative stress induced by sonodynamic therapy to the calcium homeostasis imbalance enhances macrophage infiltration"
Since you have shown the up regulation of 2 markers (1 M1 and 1 M2) this would be acceptable. Your future direction could be to characterize the macrophage lineage/polarization.
Best wishes
This manuscript is a resubmission of an earlier submission. The following is a list of the peer review reports and author responses from that submission.
Round 1
Reviewer 1 Report
The authors provide an assessment of the use of sonodynamic therapy on Calcium channel homeostasis with pharmacologic synergistic treatment. The authors use a Piezo1 antagonist and an agonist to demonstrate the importance of Ca++ regulation for glioma viability in this model. The premise of the manuscript is that Piezo1 antagonism can improve an antitumor effect by recruiting macrophages and disrupting Ca-lipid complexes and increase ROS. The authors have a good scientific method and demonstrate their hypotheses well. A few points of feedback.
- The English in the manuscript is poor, and would benefit from a editor to review the syntax. Examples:
- abstract "we explored more effect of SDT"
- abstract "under SDT later 43 period"
- introduction "with deepen study of the interactions between cells"
- The diagram of Piezo1 interactions with SDT is not easy to follow. There is too much being displayed, and it is hard to follow for someone who is not an expert in mechxnosensitive Ca channels.
- It is good that the authors used two separate glioma cell lines. Did they notice any big differences in Ca regulation between the two cell lines? Is there a difference between adherent cell lines and neurospheres?
Author Response
We thank you very much in advance for your time and contribution. Please see the attachment (point-by-point responses to the reviewer 1 comments).

Reviewer 2 Report
Dear authors,
I have been asked to review your article entitled “Contribution of oxidative stress induced by sonodynamic ther- 2 apy to the calcium homeostasis imbalance enhances M1 macro- 3 phage polarization in glioma cells”, where you examined the hypothesis that early application of mechanosensitive Ca2+ channel antagonist Piezo1 (GsMTx4) could better promote the dissociation and polymerization of Ca2+-lipid complex and further increase the level of oxidative stress, leading to better anti-tumor effect under the treatment of sonodynamic therapy SDT for brain gliomas
The hypothesis is interesting and well-sounded. I really appreciate your effort that you invested in producing such a demanding paper.
However, it is unclear to me are the results you have gained originated from mice or human model?
The entire Methods section is far too long and a bit confusing, and it needs to be shortened and rearranged to allow better insight into the matter.
The Results are also presented in an unclear manner and not executed well. They are not designed as self-explanatory, and therefore belong more to the Discussion. Please shorten this section.
I would also suggest including a few sentences concerned with the study limitation at the end of the Discussion, as well as engaging a nature English speaker to improve the language flow.
Lastly, I believe that the term glioblastoma multiforme is out of the use after new WHO CNS tumor classification has been introduced a couple years ago. Therefore it needs to be replaced with the term glioma of the highest grade or grade IV glioma.
Author Response
We thank you very much in advance for your time and contribution. Please see the attachment (point-by-point responses to the reviewer 2 comments).
